# The Effect of Radial-Shear Rolling Deformation Processing on the Structure and Properties of Zr-2.5Nb Alloy

**DOI:** 10.3390/ma16103873

**Published:** 2023-05-21

**Authors:** Kirill Ozhmegov, Anna Kawalek, Abdrakhman Naizabekov, Evgeniy Panin, Nikita Lutchenko, Sanzhar Sultanbekov, Medet Magzhanov, Alexandr Arbuz

**Affiliations:** 1Faculty of Production Engineering and Materials Technology, Częstochowa University of Technology, ul. J.H. Dąbrowskiego 69, 42-201 Częstochowa, Poland; kvozhmegov@wp.pl (K.O.); kawalek.anna@wip.pcz.pl (A.K.); 2Rudny Industrial Institute, 50 Let Oktyabrya Street 38, Rudny 111500, Kazakhstan; 3Metal Forming Department, Karaganda State Industrial University, 30 Republic Ave., Temirtau 101400, Kazakhstan; cooper802@mail.ru; 4Core Facilities Department, AEO Nazarbayev University, 53 Kabanbay Batyr Ave., Nur-Sultan 010000, Kazakhstan; nikita.lutchenko@nu.edu.kz; 5Mechanical Engineering Department, Abylkas Saginov Karaganda Technical University, 56 Nursultan Nazarbayev Ave., Karaganda 100027, Kazakhstan; sanzhar.stb@gmail.com; 6JSC Volkovgeologiya (NAC Kazatomprom), Bogenbay Batyr Str. 168, Almaty 050012, Kazakhstan; mr.medet@outlook.com

**Keywords:** radial-shearing roll, rheology, dilatometry, ultrafine-grained structure, zirconium alloy, gradient structure, severe plastic deformation, numerical simulation

## Abstract

The rheological properties of the Zr-2.5Nb alloy by the strain rate range of 0.5–15 s^−1^ and by the temperature range of 20–770 °C was studied. The dilatometric method for phase states temperature ranges was experimentally determined. A material properties database for computer FEM simulation regards the indicated temperature-velocity ranges were created. Using this database and DEFORM-3D FEM-softpack, the radial shear rolling complex process numerical simulation was carried out. The contributed conditions for the ultrafine-grained state alloy structure refinement were determined. Based on the simulation results, a full-scale experiment of Zr-2.5Nb rod rolling a on a radial-shear rolling mill RSP-14/40 was carried out. It takes in seven passes from a diameter of 37–20 mm with a total diameter reduction ε = 85%. According to this case simulation data, the total equivalent strain in the most processed peripheral zone 27.5 mm/mm was reached. Due to the complex vortex metal flow, the equivalent strain over the section distribution was uneven with a gradient reducing towards the axial zone. This fact should have a deep effect on the structure change. Changes and structure gradient by sample section EBSD mapping with 2 mm resolution were studied. The microhardness section gradient by the HV 0.5 method was also studied. The axial and central zones of the sample by the TEM method were studied. The rod section structure has an expressed gradient from the formed equiaxed ultrafine-grained (UFG) structure on a few outer millimeters of the peripheral section to the elongated rolling texture in the center of the bar. The work shows the possibility of processing with the gradient structure obtaining and enhanced properties for the Zr-2.5Nb alloy, and a database for this alloy FEM numerical simulations are also presents.

## 1. Introduction

The nuclear energy industry development is also associated with the extension of the service life of existing CANDU channel type reactors. At the same time, more energy-intensive and competitive nuclear setup of this type are being developed [1,2,3]. The most important structural elements of channel reactors are pressure pipes [4,5,6,7], the integrity of its the normal operation and safety of nuclear power plants determines. The design life of channel reactors is 30 years [6]; however, pressure pipes made of zirconium alloys operate less than the design period [7,8,9]. Cases of depressurization were observed for CANDU pressure pipes by various sources [10,11,12]. Therefore, the task of improving the material of pressure pipes for channel reactors is extremely important. 

The functional properties of zirconium alloys are determined by its composition and structure. In the reactor core, zirconium components have significant structural and phase changes, leading to a change in mechanical properties, corrosion, hydrogenation, shape change (radiation creep and growth), and interaction with fuel fission products [11].

Knowing the regularities and laws of such changes depending on the composition and initial structure of zirconium alloys facilitates the choice of material for a specific application as a fuel element cladding, pressure pipe, or parts of the fuel assembly frame.

The Zr-2.5Nb alloy is used for pressure pipes of CANDU channel reactors [11]. The manufacturing technology of channel pipes includes the operations of hot forging, hot pressing, and cold pilger rolling [12]. The average grain size, determined along any line at right angles to the surface of the finished product, should not exceed 15 micrometers, and there should not be grains larger than 35 micrometers, that is, the average grain size should be the average distance between grain boundaries [13,14]. Available industrial technologies for manufacturing products from zirconium alloys provide obtaining equiaxed grains with a size of 1–5 μm [15].

An increase in the fuel cycle of fuel assemblies (FA) is possible due to the formation of the ultrafine-grained (UFG) structure of the metal of the structural elements of the fuel assemblies. This structure will improve performance [16], including mechanical characteristics [17] and corrosion resistance [18].

The structure and properties changing by severe plastic deformation (SPD) study of metallic materials, including Zr/Nb-based alloys, are also relevant. The structure formation regularities by such basic SPD processes equal-channel angular pressing (ECAP) for Zr-2.5%Nb alloy presented by [19,20] or as high-pressure torsion (HPT) by [21] are known and studied. They were studied not only for mechanical properties changes but for corrosion resistance [22], cracking resistance [23], fatigue strength [24], and other properties [25]. These sources can give general information not only on the study relevance and zirconium properties changes but also on experimental tech and used equipment features. However good the scientific results, these processes are lab-based and cannot be applied for industrial use.

The UFG state means the grain size range between around 1 µm and less than 100 nm. This fact means greater grain boundaries in material volume than the conventional grained state with 50–80 µm grain size. This fact means many more grain boundaries impact the macro properties of the whole material. It could be mechanical properties, some physical properties, and its unusual combinations, irradiation resistance, etc. The UFG grains should have large crystallographic misorientation angles and have an equiaxial shape. One of most important outputs are the high strength within high plasticity combination All of this UFG phenomena mechanism is described in detail by R. Valiev [26], T. Langdon, and Z. Horita [27] in their works or in some other sources like [28,29]. One of the main aims of this research is to obtain a similar to that described in the UFG classic work [26] structure type on an example of Zr-2.5Nb alloy. 

Despite the significant interest in zirconium alloys in regards to improving service properties by obtaining ultrafine-grained structures, there is not enough published on the methods of forming an ultrafine-grained or nanocrystalline structure when processing products from zirconium alloys. To date, there are no data on industrial SPD processes and modes of subsequent heat treatments that ensure the formation of a structure with a less than 1 µm grain size on finished products. In particular, the effect of SPD modes and heat treatments on structure formation (in the entire volume of the test sample) and mechanical and corrosion properties of the industrial Zr-2.5Nb alloy has not been sufficiently studied. 

Metal processing by radial shear rolling mills is the one of likely ways to obtain UFG structures [30]. The radial-shear rolling (RSR) deformation zone stress–strain state scheme is close to all-round compression with big shear strains. The radial-shear rolling scheme and its internal metal flow features shown Figure 1 is taken from [30]. These conditions are highly likely for UFG structure transformation. 

The deformation non-monotonicity and metal flow turbulence is the main radial-shear rolling feature. This nice phenomenon by the workpiece caused different zones trajectory-speed characteristics of plastic flow differences [31]. These features are shown in the Figure 1 scheme. This is why the most intense shear deformations in the sliding lines intersection zone are localized. Each small trajectory-oriented element of the outer layer is subjected to compression strain along the workpiece radius, compression strain in the flow direction along the helical trajectory, and stretching strain across the helical trajectory. The values and vector of all processes have a gradient along the workpiece radius. Metal flow currents have no sharp border and the fact that the additional grain refinement conditions are added [31].

The axial workpiece zone metal flow currents look like a normal pressing process by the all-round 3rolls pressure for workpiece causation. The metal is simply extruded from the central zone. The strain rate is also decreased, and the metal flow direction and the workpiece axis are matched. The metal structure should to stretched to become the texture. All of this is theory by S. Galkin and described in detail in [31]. 

This work goal is the evaluation of the radial shear rolling process for the Zr-2.5%Nb zirconium alloy structure refinement applicability to increase its performance.

## 2. Materials, Methods, and Equipment

One of the common zirconium-based alloys, the Zr-2.5% Nb alloy, was chosen for this research. This alloy is used as a material for CANDU pressure pipes, and can also be used as a nuclear fuel cladding tube material and its end plugs. There are few works on the severe plastic deformation of this alloy, and its radial shear rolling has not yet been carried out.

To understand the concept of metal flow and its plasticity features in regards to the temperature and speed conditions of the Zr-2.5% Nb alloy radial shear rolling, and taking into account the plastic deformation thermal effect, a plastometric study was carried out. The plastometric tests by the method of uniaxial compression of cylindrical specimens with a 10 mm working zone using the strain rates of 0.5–15 s^−1^ and the temperature range of 20–770 °C were implemented. Cylindrical samples Ø10 × 12 mm were cut from an annealed bar Ø37 mm. The grain size corresponded to six points according to ASTM E112. The ratio between the length and diameter of the sample is h/d = 1.2. An increase in this ratio on zirconium alloys is undesirable since it leads to sample collapse and loss of stability during upsetting.

The continuous loading test conditions by the Gleeble 3800 plastometric unit by the «Pocket Jaw» module were carried out. The Gleeble 3800 setup makes it possible to simulate the various metal-forming processes conditions [32,33,34]. The temperature accuracy is about ±1 °C. 

The test temperature chromel-copel thermocouple wire in the sample central part was controlled. It was connected by the Gleeble 3800 “Therwocouple welder” kit. The thin graphite-based gaskets as a lubricant for the tests were used. The ISO-T model working heads of the test instrument were additionally lubricated by OKS255 graphite grease after each test.

The structural transformations as applied to rolling conditions to study the dilatometric tests were carried out. Cylindrical specimens Ø5 × 10 mm were cut from an annealed bar Ø37 mm. The grain size corresponded to six points according to ASTM E112. These studies were carried out using a deformation dilatometer DIL805 A/D. The samples were heated to a temperature of T = 700–800 °C at a heating rate of 10–20 °C/min. The strain rate was έ = 0.5 s^−1^, and the strain degree ε = 0.5. The cooling rate after deformation was 0.5 °C/s. 

The Deform-3D program (SFTC, Columbus, OH, USA) for computer simulation by the finite element method (FEM) was carried out. The RSP-14/40 rolling mill from Częstochowa University of Technology [29,30] for real technical parameters for the radial shear rolling basic FEM-model creation was used. This rolling mill will be described in detail. The original 37 mm diameter workpiece 150 mm in length with the reductions indicated in Table 1 was rolled. Based on previous experience with this mill using other materials and its computer simulations, the reductions were determined [30,35,36,37]. The workpiece material is Zr-2.5%Nb alloy. This material is not presented in the Deform database, and due to it, the plastometric studies results as a new database library were imported. As a result, a new library of the studied material was created for the Deform program.

The heating temperature of 530 °C was chosen as the maximum possible to exclude the α→α + β phase transition; the roll speed of the RSP-10/30 mill was a nominal 100 rpm. The billet and rolls contact zone friction coefficient of 0.7 were taken as the Deform recommended value. The KOMPAS-3D (by Askon) CAD-softpack for the rolls geometry 3D model *.STL drawing was used. The workpiece material as an elastic-plastic type was chosen. The rolls as rigid bodies type was modeled.

The computer simulation results verification on the Częstochowa University of Technology by the RSP-14/40 rolling mill (ZAO “ISTOK ML”, Moscow, Russia) was carried out. The RSP-14/40 rolling mill at NUST MiSIS was developed. The rolling setup for the hot deformation of round bars made from unusual materials, low-plasticity, compacted powder, and casted bars was developed. The workpiece’s bar diameter range is 40–8 mm. The main unit setup has a special rigid design three-roll stand rolls with specially calibrated rolls of 90 mm in diameter. The mill has two different diameters roll sets for different ranges of rolling bars. Large roll sets can roll 40–19 mm diameter bars. However, in practice, the rolling mill does not roll 40 mm, and the minimum possible diameter turned out to be 37 mm. Therefore, just such a rolling route was chosen in order to roll the billet to the maximum deformation from the one heating.

The same rolling mill for equiaxed 300–700 nm ultrafine-grained structure obtained for austenitic stainless steel AISI-321 and Zr-1%Nb alloys was previously successfully used [35,36,37]. It has wide adjustments, high stand rigidity, and ease of operation. The RSP-14/40 rolling mill is shown in Figure 2.

For experimental rolling, a 40 mm initial diameter Zr-2.5%Nb alloy rod was used. It was prepared by hot (650 °C) pressing with drawing µ = 25. Then the bar was machined to a diameter of 37 mm. The structure of the original bar was recrystallized with a grain size of six points according to ASTM E112.

The original pressed rod mechanical properties, according to the manufacturer’s data, are as follows: tensile strength = 520 MPa, yield strength = 390 MPa, elongation = 17.5%. To assess the mechanical properties by the HV 0.5 method, a Shimadzu HMV-G31ST microhardness tester with a Vickers indenter tip was used. Microhardness was measured with load 5 (4.903) N and a dwell time of 5 s with the step of 0.25 mm. Microhardness was measured on smooth etched specimens after EBSD. The use of electrolytic etching guarantees the absence of a surface layer deformed by the abrasive and the most accurate measurements. Each point on the microhardness graphs is the average of five measurements. The microhardness was chosen due to the presence of a gradient structure, which makes it difficult to correctly use tensile tests.

The heating temperature was set at 530 °C, and the heating of the initial rod with a diameter of 37 mm was carried out in a preheated muffle furnace for 40 min. An infrared thermal imaging camera aimed at the deformation zone was used to control the temperature regime.

Sample cutting for all types of subsequent sample preparation was performed on a precision cutting machine Brilliant-220 (QATM) with intensive water cooling and a cutting speed of 15 µm/s to minimize deformation-temperature damage to the structure. Coarse-grained cut-off wheels were used at 500 RPM. Grinding and polishing with a Sapphire-520 machine (QATM) were carried out. The scheme of cutting the bars for sample preparation and the places of analysis are schematically shown in Figure 3.

The microstructure by transmission electron microscopy (TEM) using a bright field and electron diffraction mode on the JEM-1400 Plus microscope (Jeol Ltd., Tokyo, Japan) at an accelerating voltage of 120 kV and magnification ×8000–×35,000 was studied. The structure gradient was studied on a CrossBeam-540 scanning electron microscope (SEM) (Carl Zeiss, Oberkochen, Germany) at 20 kV using a NordlyssNano EBSD detector (Oxford Instruments, Abingdon, UK).

The EBSD mapping using the following conditions was obtained: accelerating voltage 30 kV; work distance (distance from column) 10 mm with sample tilt 70° (pretilted holder); magnification 8K; map resolution 318 × 234 pixels or 14 × 10 µm; step size (pixel size) 45.1 nm; scanning time 30 min.

The TEM sample preparation by the jet thinning method was carried out. A 10 mm long sample from the central part was cut. Then it was half cut and several axial region longitudinal thin (0.3 mm) sections were obtained. Thus, the better radial shear rolling microstructure characterization longitudinal section was used.

Then, by electrolytic jet thinning, the final TEM samples were prepared on a TenuPol-5 unit (Struers, Copenhagen, Denmark) A3 electrolyte (600 mL Methanol, 360 mL Butylcellosolve, 60 mL perchloric acid). Some of the samples were used to test sample preparation modes. For electrolytic polishing of TEM samples, longitudinal plates 0.3 mm thick were cut from the central part of the rod, and 3 mm discs were knocked out of them with a disc punch tool from Gatan (USA), then these blanks were thinned to form a hole at a voltage of 20 V.

For SEM/EBSD, thicker (2 mm) plates were also cut. The cutting scheme is shown in detail in the experimental section. Sample preparation of zirconium samples was also carried out by electrolytic polishing using a LectroPol-5 setup (Struers, Denmark) for SEM/EBSD in a solution of the already mentioned electrolyte A3. 

## 3. Results and Discussion

### 3.1. The Zr-2.5%Nb Alloy Rheological Properties

Based on Gleeble 3800 plastometer plastometric tests results, the Zr-2.5%Nb alloy stress–strain flow curves graphs regarding the 0.5–15 s^−1^ strain rates range and 20–770 °C temperature range were plotted. To construct each curve, three tests were carried out. A total of 30 tests were carried out. The flow curves are shown in Figure 4.

It can be observed that the deformation resistance decreased by approximately 80% as a result of the change in temperature from 20 °C to 770 °C, whereas the opposite phenomenon occurred due to an increase in the rate of deformation from 0.5 to 15 s^−1^. At a temperature of 20 °C, the increase in deformation resistance is no more than 5%. At 770 °C, the increase in resistance to deformation is approximately ~10%. This difference in the influence of the strain rate can be explained by the thermal effect of plastic deformation. At temperatures close to room temperature, it is higher than at higher deformation temperatures. 

The type of flow curves depends on the different temperature and strain rate combinations as evidenced by the following points. The increase in the number of slip systems involved in deformation combined with the ongoing processes of dynamic recovery leads to the hardening coefficient showing a marked decrease between 350–500 °C and 0.3–0.4 strain rate values. 

Depending on the strain rate, the 770 °C flow curves are dome-shaped with a maximum deformation resistance at corresponding values of 0.15–0.30. Further, the deformation resistance reaches a steady state at a point slightly below 200 MPa. This phenomenon is typical for HCP metals and alloys. These materials are distinguished not only by a significant thermal effect under conditions of cold and warm deformation at high rates but also by an expressed anisotropy of properties with the textural inhomogeneity.

### 3.2. The Dilatometric Properties of Zr-2.5%Nb Alloy

Six dilatometric experiments were carried out where during the cooling process after deformation, the change in the length of the samples was recorded. During the tests, temperature regions were noted in which there was a change in the nature of the curves in the “temperature-length” diagram. An analysis of the results of dilatometric studies of the Zr-2.5%Nb alloy showed that the influence of the heating and testing temperature, the cooling rate after heating does not clearly manifest itself in the change in the elongation of the samples during cooling after deformation. It was noted that with a drop in temperature, structural changes occurred in the samples, which were reflected in an increase or decrease in their length. The determination of the expansion coefficient made it possible to distinguish three temperature regions of the alloy under study, which clearly differ from each other in the slope of the curve sections on the “temperature-length” diagram (Figure 5). When the samples were cooled from T = 770 °C to T = 650 °C, the coefficient of expansion of the Zr-2.5%Nb alloy ranged from –0.4 to –0.2. The negative value of the expansion coefficient is explained by the fact that the metal contracts during cooling. However, with further cooling to T = 530 °C, the expansion coefficient is already –0.03, that is, the process of narrowing the samples slows down. Most likely, in the temperature range of 530–650 °C, the rearrangement of the lattice of the Zr-2.5% Nb alloy from bcc to hcp ends [38]. Since in the hcp lattice, the ratio of the lengths of the faces perpendicular to each other is greater than in the bcc lattice, the expansion in the alloy under study during cooling can be explained by this rearrangement. With further cooling to T = 200 °C, the softening coefficient was in the range from −0.03 to +0.03.

### 3.3. The Zr-2.5%Nb Alloy Radial-Shear Rolling Computer Simulation

The Deform software (SFTC) for FEM simulation was used. The initial 37 mm diameter billet and 150 mm length according to Table 1 specified compressions to 20 mm diameter was rolled. based on a previous study of Zr-1Nb alloy [30] compressions bypass were determined. 

To analyze the metal processing level during deformation, the parameter “equivalent strain” is usually used. Since radial-shear rolling is a cross-type of rolling, it is advisable to study the equivalent strain in the cross-section of the workpiece—it will allow evaluating not only the numerical values of the parameter but also the nature of its distribution over the cross-section during deformation. 

When analyzing the equivalent strain (Figure 6), it was found that the distribution of this parameter has a ring-type—in all cross sections there are clear ring zones of strain development. At first pass, when compression was 2 mm per pass, the difference of strain values between the center and surface has a smooth gradient view. After the last pass with a 2 mm compression (pass 4), in the axial zone the strain level is approximately 9.3, in the surface zone, where the shear deformation maximum effect is detected, the approximate strain level is 15 (Figure 7).

When the compression level was increased up to 3 mm per pass, this led to an increase in the strain level difference between the center and surface. After two passes (pass 6) in the axial zone, the strain level is approximately 17.5, in the surface zone, where the maximum shear deformation effect is detected, the strain level is approximately 23.5. 

At the last 7th pass, the billet was deformed from 23 mm to 20 mm. But for these roll construction, such deformation mode led to intensive processing over the whole cross-section. Further deformation at this mill is impossible as rolls start to touch each other. In this case, the metal during rolling obtained the deformation of compression mainly, and it is more like an extrusion process. The result of this deformation stage is a sharp decrease in the strain values difference in the cross-section. Therefore, in the axial zone, the strain level is approximately 25.5, and in the surface zone, where the maximum effect of shear deformation is observed, the strain level is approximately 27.5.

Based on the computer simulation results, recommendations were developed for rolling bars from the Zr-2.5%Nb alloy to ensure a high workout of the structure without metal crushing.

### 3.4. The Zr-2.5%Nb Alloy Radial Shear Rolling and Its Microstructure Changes

The experimental rolling by the RSP-14/40 rolling mill at Częstochowa University of Technology was carried out regarding the setup limiting mechanical and technological conditions. During the experiment, there were roll jamming in several cases. After each rolling pass, the bar was quickly removed and placed in a furnace for saving heat. The rolling was conducted for a 1.5–3 mm diameter reduction per pass to the 20 mm final diameter. The 20 mm final workpiece diameter is the technological limit for large rolls. The smaller diameter roll set replacement and installing requires a lot of time. For this reason, the smaller roll diameter rolling by one heating is not possible. 

A short-term 50–150 °C surface temperature increase was detected as the plastic deformation thermal effect. The final rolled workpiece was air cooled.

Comparison of the shape change of the rolled experimental bar with the previously obtained FEM model shows good convergence of the results and the same shape of the front end with depression due to the vortex flow of metal in the bar. The comparison is shown in Figure 8. In addition, the annular zones of strain distribution over the cross-section of the rod correlate with the resulting structure gradient. The greater the deformation value (during simulation), the smaller the grain (during a laboratory experiment). In addition, the results of verification should include measurements of the thermal effect from the surface of the bar with the data obtained during the simulation. Model c predicts the temperature on the bar surface with an error of no more than 10%.

The study of the microstructure of TEM is shown in Figure 9 and Figure 10. TEM images of the peripheral zone (Figure 9, left) show equiaxed dislocation-saturated fine grains ranging in size from 700 to 1100 nm. A more detailed picture of the complex dislocation structure is shown in Figure 11. The electron diffraction pattern shows the oriented texture absence and presents the high-angle grain boundaries dominance. This structure type is the best for reaching high property levels.

The axial (central) zone structure presented in Figure 9 (right) has also significance. Instead of large randomly orientated grains, after the radial shear rolling processing, the mixture of long narrow elongated strongly deformed grains was formed. Sharp edge straight parallel boundaries are visible clearly and show one direction high-deformed texture. The grains’ orientation here is the same and correspond to the rolling direction. 

To characterize the gradient of the structure observed on the TEM, EBSD mapping was used with a step of 2 mm. The original EBSD images and its misorienation data you can find in Appendix A. The captured maps were recognized and statistically processed. The main indicators taken from the maps are the average grain size (bars) and the average size ratio (red graph). The average grain size does not sufficiently characterize the structure gradient, whereas the grain size ratio provides much more information about this. The grain aspect ratio was taken by dividing the smaller side by the longer side which will equal the 0…1 range. Close to 1 value means the grain is close to the circle shape, and close to 0 value means the grain is close to the strip shape. The average value changes from 0.5 in the peripheral zone to 0.3 in the axial zone. It is safe to say that the outer 2–3 mm of the 10 mm rod radius has an equiaxed UFG structure with high-angle boundaries. Then, as we approach the center of the bar, the predominant orientation of the grain changes, and the shape becomes more and more elongated. At the same time, on large maps, zones of a texture similar to that shown in Figure 9 (right) are interspersed with individual large grains that look like recrystallized or small clusters of small relatively equiaxed grains. This can be seen from the EBSD thumbnails in Figure 11.

An important conclusion should also be made that under the conditions of the formation of a non-equilibrium gradient structure, the main characterization method should be considered detailed EBSD mapping at different scales, since the TEM analysis is too local and, as can be seen from the correlation of the TEM and EBSD results for the axial zone, the results may not match exactly. TEM should be considered as an auxiliary method aimed at a more detailed study of one or another of the discovered types of structure.

Another method for characterizing the gradient structure is the measurement of microhardness over the section shown in Figure 12. Analysis of the results confirms the gradient nature of the formation of the structure over the cross-section of the bar.

The structure of bars Ø20 mm made of Zr-2.5%Nb alloy is not typical of the structure obtained by traditional technology using pilger rolling mills.

In the near-surface layers (at a distance of 10 mm), a predominantly recrystallized structure with an average grain size of 0.75 μm is noted. Such a structure was obtained as a result of significant local shear deformations at the temperature of the beginning of rolling T~530 °C, taking into account the thermal effect of plastic deformation up to ∆T~150 °C and cooling in the air after rolling. In fact, the temperature at the time of rolling on the metal surface increased to T~680 °C for a short time (no more than 10 s), and then, due to the small cross-section of the metal, it cooled in air to T = 300 °C in 35 s. The average cooling rate is Vcool = 10 °C/s according to the results of pyrometric studies. In view of the significant local work hardening as a result of radial shear rolling, a short-term thermal effect was sufficient for the formation of a UFG recrystallized structure in the process of dynamic recrystallization.

At a distance of 7 mm from the center, a predominantly recrystallized structure is also observed; however, the average grain size is at the level of 1.0 μm. Such a structural state is probably associated with an increase in the time of thermal influence due to a decrease in the cooling rate.

At a distance of 4 mm from the center and in the center, the structure acquires a state after hot deformation processing of the Zr-2.5% Nb alloy by hot pressing at T = 650–700 °C, in which the turbulent movement of the metal predominates in the presence of local areas with tensile stresses. Some single grains are characterized by an elongated structure. The pressing process is longer in time and is characterized by lower strain rates.

The presence in the metal of a temperature field T~500–680 °C and an unevenly distributed stress–strain state over the cross-section of the rod explains its inhomogeneous structural state. The decrease in the hardness value in the direction from the center to the periphery of the section of the bar Ø10 mm is associated with the degree of recrystallization. According to the results of the study, in the peripheral regions, the degree of recrystallization is higher than in the center, regardless of the grain size.

## 4. Conclusions

Based on the results of plastometric and dilatometric studies, using computer simulation to argue the thermomechanical parameters of radial shear rolling of zirconium Zr-2.5%Nb alloy bars, the calculations were carried out. 

It is noted that the rolling temperature should not exceed T = 530 °C in order for the deformation process to take place advantageously in the single-phase α-region. It is advisable to give single reductions per pass in the range from 10 to 25% in order to provide an ultrafine-grained structure in the near-surface layers and prevent metal destruction.

The verification on the RSP-14/40 radial-shear rolling mill was performed. The round bar by the 37 mm→20 mm route with a total diameter reduction of about ε = 85% was processed. The equiaxed ultrafine-grained 700–800 nm structure was reached. The formed structure over the cross-section of the sample has a gradient character. The zone occupied by the UFG structure formed on the periphery of the sample, whereas in the center an oriented rolling texture with an admixture of 1.0–1.5 μm grains was formed. 

The research demonstrated the radial shear rolling applicability for Zr-2.5%Nb alloy deformation for the UFG structure produce. It is suitable to consider the radial shear rolling implementation mainly towards the end of the technological cycle of manufacturing products in order to maintain the achieved effect in finished products. The laboratory experiment and the computer simulation also result in high convergence demonstrated.

## Figures and Tables

**Figure 1 materials-16-03873-f001:**
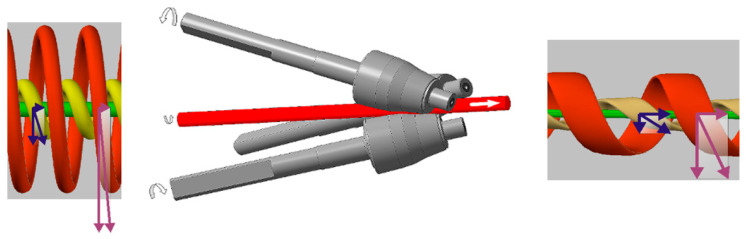
The radial-shear rolling scheme and metal flow features by the different workpiece zones.

**Figure 2 materials-16-03873-f002:**
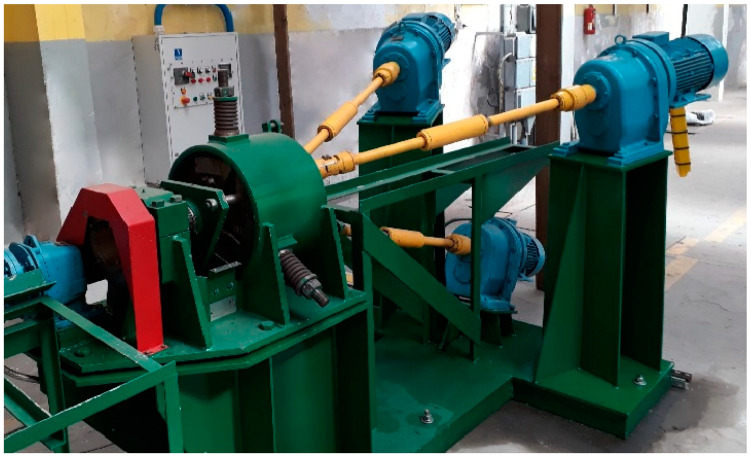
The RSP-14/40 radial shear rolling mill.

**Figure 3 materials-16-03873-f003:**
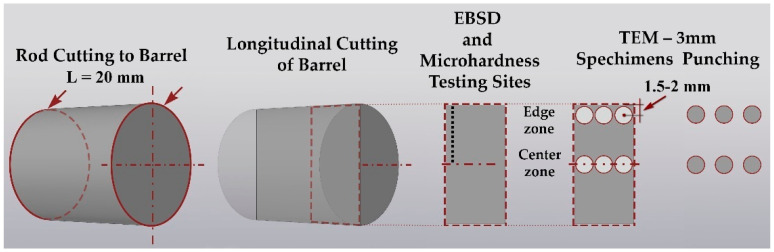
Sample cutting scheme for various characterization methods.

**Figure 4 materials-16-03873-f004:**
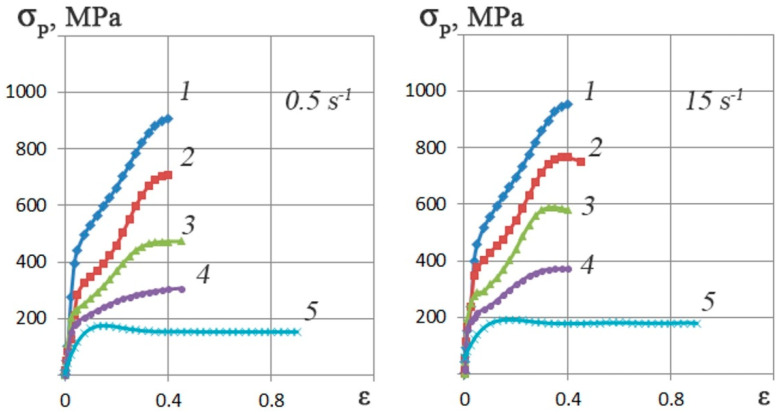
The Zr-2.5%Nb alloy by the Gleeble 3800 plastometer flow curves for the compression method in the strain rate range of 0.5–15 s^−1^ at temperatures: 1–20 °C; 2–200 °C; 3–350 °C; 4–500 °C; 5–770 °C.

**Figure 5 materials-16-03873-f005:**
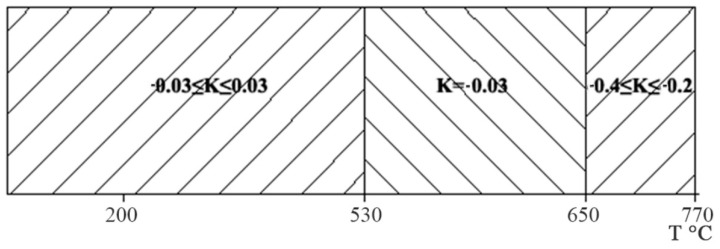
Temperature ranges of structural transformations in the Zr-2.5%Nb alloy during cooling after deformation.

**Figure 6 materials-16-03873-f006:**
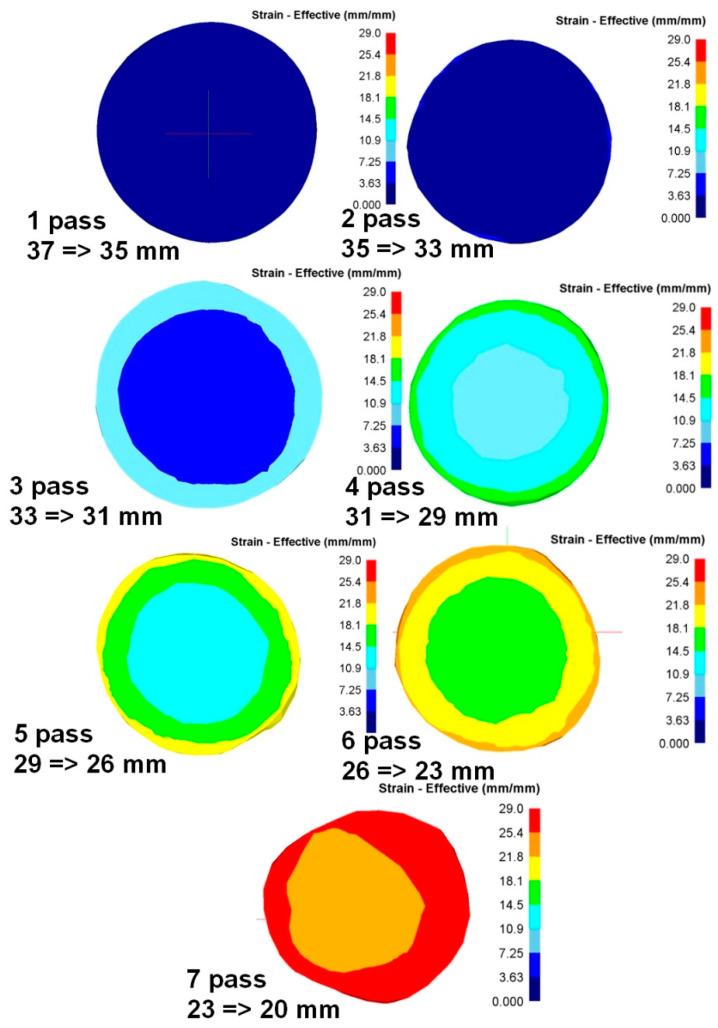
Equivalent strain evolution during radial-shear rolling seven passes on RSP-14/40 setup.

**Figure 7 materials-16-03873-f007:**
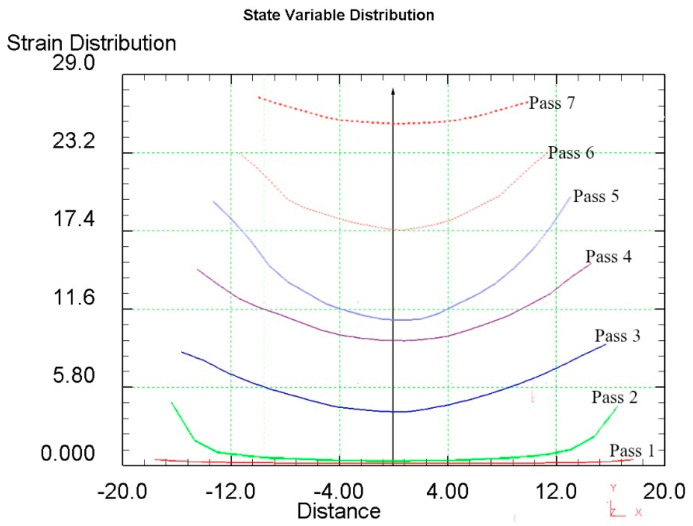
The equivalent strain distribution graph over the workpiece cross-section.

**Figure 8 materials-16-03873-f008:**
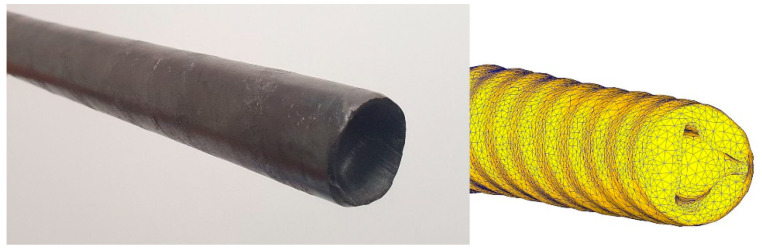
Comparison of the shape change of the rolled experimental bar with the previously obtained FEM model.

**Figure 9 materials-16-03873-f009:**
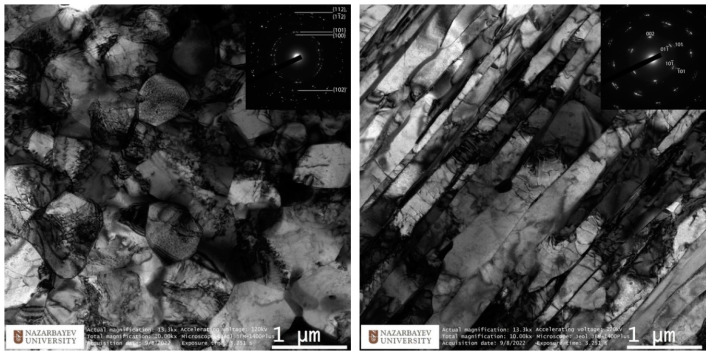
TEM-characterization of final sample peripheral zone structure (**left**) and axial (central) zone structure (**right**).

**Figure 10 materials-16-03873-f010:**
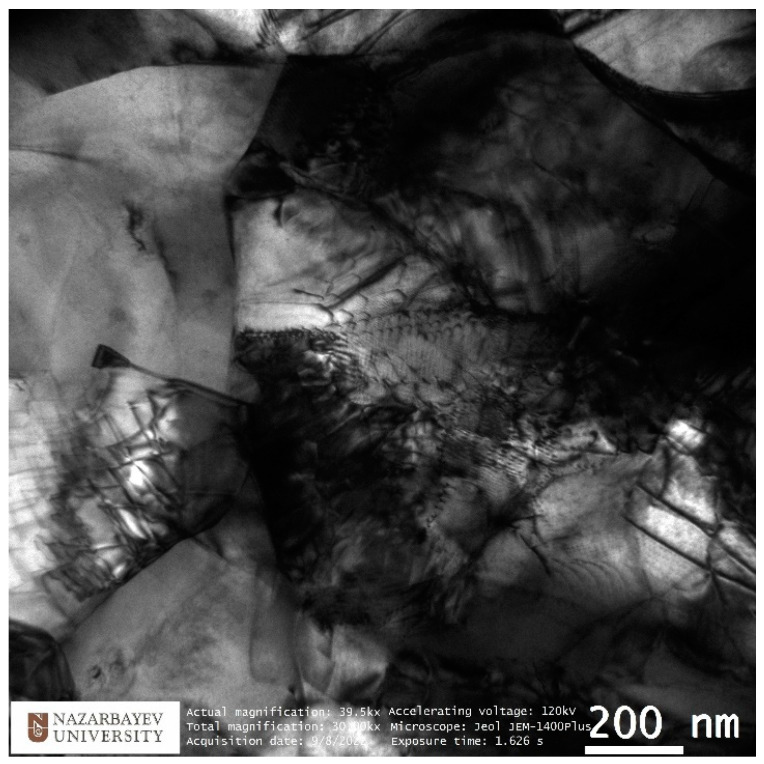
TEM-characterization of peripheral zone fine structure peculiarities.

**Figure 11 materials-16-03873-f011:**
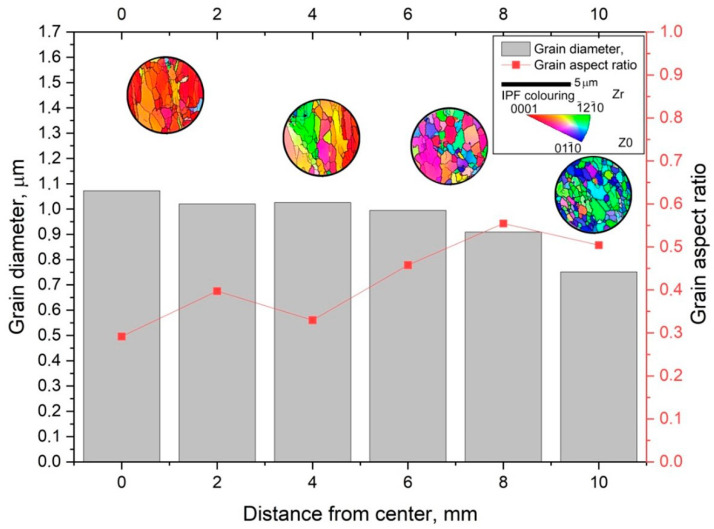
Diagram of the evolution of the structure from the periphery to the center of a Zr-2.5%Nb alloy bar after rolling on an RSP-14/40 rolling mill.

**Figure 12 materials-16-03873-f012:**
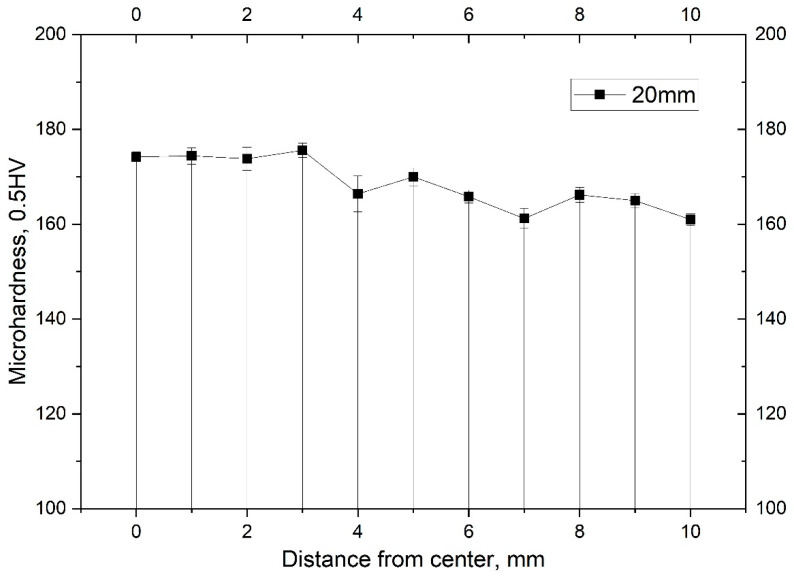
The results of the study of microhardness HV 0.5 from the periphery to the center of the Zr-2.5%Nb alloy bar after rolling on the RSP-14/40 rolling mill.

**Table 1 materials-16-03873-t001:** Rolling parameters of Zr-2.5%Nb bar on the RSP-14/40 radial-shear rolling mill.

Pass Number	D_0_, mm	D_1_, mm	Reduction, mm
1	37	35	2
2	35	33	2
3	33	31	2
4	31	29	2
5	29	26	3
6	26	23	3
7	23	20	3

## Data Availability

Not applicable.

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
