# Peer review of "The Effect of Radial-Shear Rolling Deformation Processing on the Structure and Properties of Zr-2.5Nb Alloy"

_materials, 2023, doi:10.3390/ma16103873_

Round 1
Reviewer 1 Report
In this paper, the effect of radial-shear rolling deformation processing on the structure and properties of Zr-2.5Nb alloy was investigated. The research demonstrated the radial shear rolling applicability for Zr-2.5%Nb alloy deformation for the UFG structure produce. In my opinion, several aspects should be modified before publication.
Some comments are given as follows:
1) Moderate English changes must be required to increase the readability. Such as, line 48, the task of improving the properties?? material of pressure pipes….
2) In the revised paper, the relationship between the microstructure and mechanical properties should be discussed.
3) In figure 12, the grain aspect ratio is usually more than 1, please modify the figure.
4) In Figures 12 and 13, it seems the finer grain size, the lower hardness, why??
Author Response
Dear Reviewer,
Thank you so much for our paper consideration and for your comments.
The answer for 1st point
1) Moderate English changes must be required to increase the readability. Such as, line 48, the task of improving the properties?? material of pressure pipes….
Thank you for this comment. The text was proofread and some English changes were made.
The noted line 48 issue: the general task was to zirconium-made parts properties improving. It is correct. We have improved microstructural state properties by Zr-2.5%Nb alloy. It is also correct. The Zr-2.5%Nb alloy is the key material for CANDU-type reactor pressure pipes. All of these points are still correct.
The answer for 3rd point
3) In figure 12, the grain aspect ratio is usually more than 1, please modify the figure.
The grain aspect ratio was taken by dividing the smaller side by the longer side which anyway will equal the 0…1 range. Close to 1 value means the grain is close to the circle shape, and close to 0 value means the grain is close to the strip shape. Something like a roundness percentage 0...1 range.
This clarification information should was added to the text earlier. We added it now and highlighted it in green. We consider this method more informative and more general and we can use the same scale bar for all of our samples for best comparisons
The combined answer for 2nd and 4th points
2) In the revised paper, the relationship between the microstructure and mechanical properties should be discussed.
4) In Figures 12 and 13, it seems the finer grain size, the lower hardness, why??
The structure of Zr-2.5%Nb alloy Ø20 mm bars is not typical of the structure obtained by traditional technology using pilger rolling mills.
In the near-surface layers (at a distance of 10 mm), a predominantly changed structure with an 0.75 μm average grain size is noted. Such a structure was obtained as a result of significant local shear deformations at the temperature of the beginning of rolling Т~530 °С, taking into account the thermal effect of plastic deformation up to ∆Т~150 °С and cooling in the air after rolling. In fact, the temperature at the time of rolling on the metal surface increased to T ~ 680 °C for a short time (no more than 10 s), and then, due to the small metal cross-section, it was cooled in the air to T = 300 °C in 35 s. The average cooling rate is Vcool = 10 °C/s according to the results of pyrometric studies. In view of the significant local work hardening as a result of radial shear rolling, a short-term thermal effect was sufficient for the formation of a UFG structure in the process of dynamic recrystallization.
At a distance of 7 mm from the center, a predominantly changed refined structure is also observed, however, the average grain size is at the level of 1.0 μm. Such a structural state is probably associated with an increase in the thermal effect time due to a decrease in the cooling rate and less deformation intensity.
At a distance of 4 mm from the center and in the center, the structure acquires a state as after hot deformation processing of the Zr-2.5% Nb alloy by hot pressing at T = 650-700 ° C, in which the turbulent movement of the metal predominates in the presence of local areas with tensile voltages. Some single grains are characterized by an elongated structure. The pressing process is longer in time and is characterized by lower strain rates.
The presence in the metal of a T ~ 500-680 °C temperature field and an unevenly distributed stress-strain state over the cross-section of the rod explains its inhomogeneous structural state. The hardness decrease in the direction from the center to the periphery is associated with the degree of recrystallization. In the peripheral regions, the recrystallization degree is higher than in the center, regardless of the grain size.
Reviewer 2 Report
Comment 1: What is the meaning of "The one of the main aims of this research are to retain the similar to [26] structure on Zr-2.5Nb alloy." in line 82? Reference 26 seems to have little relationship with the RSR method described in this paper.
Comment 2: Reference 31 does not give a detailed description of the RSR processing process. It is suggested to supplement other documents.
Comment 3: What is the structural gradient of the original bar? How is its structural gradient reflected in the simulation process?
Comment 4: What does line 282 "The model overview on the top of Figure 4 is shown." mean? Where is the specific location?
Comment 5: What parameters are simulated by finite element method? It includes material model parameters (constitutive equation), roll 3D model parameters (such as the invasion of the hydraulic tractories), friction state parameters, etc. In particular, line 145 refers to the use of Częstochowa University of Technology real technical parameters for the radial shear rolling basic FEM-model, but no reference or specific description is given.
Comment 6: The radial nonuniformity characteristics of the equivalent strain field can be observed in the equivalent strain evolution during radial-shear rolling shown in Figure 7. How does this come about?
Comment 7: Is it possible to analyze and discuss the distribution of iso-effect variation with respect to the comparison between test verification results and simulation results?
Comment 8: Line 403 "The laboratory experiment and the computer simulation result high convergence demonstrated also." What does this conclusion refer to specifically?
Comment 9: Figure 1, Figure 6 and Figure 8 lack the secondary number, and the drawings in Figure 4 and Figure 5 do not conform to the specifications of scientific papers. Figure 9-11 can be combined into one drawing.
Comment 10: The unit is missing or incorrect (for example, Table1 MM, line 181 μ≈ 25), punctuation is incorrect (for example, line 128 0.5 ÷ 15, line 129 20 ÷ 770)
Comment 11: The manuscript requires significant grammatical corrections. The reviewer understands that this may not be the authors’ native language, so I do not expect the grammar to be perfect. However, some of the major points within the manuscript are conflated by grammatical issues.
Author Response
Dear Reviewer,
thank you for your time, consideration, and comments.
Comment 1: What is the meaning of "The one of the main aims of this research are to retain the similar to [26] structure on Zr-2.5Nb alloy." in line 82? Reference 26 seems to have little relationship with the RSR method described in this paper.
Of course, the [26] have no relationship with the described RSR method. The [26] is the most cited classical work about general features of changing metals structure to nano range in Ultra-Fine Grained (UFG) state. Prof. R. Valiev made these structural changes through discrete deformation processes like Equal-Angle Pressing (ECAP) and High-Pressure Torsion (HPT). These works and their outputs are recognized as some kind of field standard. We aimed to reach the same structure by another method (RSR) of continuous deformation and of course compare our results with the best classical work. From another hand the used in [26] methods ECAP and HPT are most used around the world and most search requests by Ultra-Fine-Grained structure will be about using ECAP or HPT processes. This fact also causated such comparison.
Comment 2: Reference 31 does not give a detailed description of the RSR processing process. It is suggested to supplement other documents.
Reference 31 is the original description of this process from their inventor, prof. Galkin. This is the clearest description of the process core and mechanics and their differences from known close processes. This solo-author article also summarizes the inventor`s experience for 20 years from the first experiment. Of course, we can add some of his previous works [downthere] including the first patent №2009737 RF with poor bad-readable English translation, but we consider do not do it. Here we should reference the most readable original source. The next references in the list after noted give for other research groups' work representation.
Galkin, S.P., Goncharuk, A.V., Daeva, E.K., Mikhajlov, V.K., Romantsev, B.A. Technology for multipass screw rolling // Stal', 2003, (9), pp. 67–68
Galkin, S.P. Trajectory of deformed metal as basis for controlling the radial-shift and screw rolling //
Stal', 2004, (7), pp. 63–66
Galkin, S.P. Regulating radial-shear and screw rolling on the basis of the metal trajectory //
Steel in Translation, 2004, 34(7), pp. 57–60
Comment 3: What is the structural gradient of the original bar? How is its structural gradient reflected in the simulation process?
The original bar structure had no gradient. The original bar structure is recrystallized with 6 points by ASTM E112 grain size standard scale. The initial structure in a computer simulation was taken into account through mechanical properties, namely through the function of resistance to deformation from changes in temperature, degree, and rate of deformation.
Comment 4: What does line 282 "The model overview on the top of Figure 4 is shown." Mean? Where is the specific location?
Thank you for your attention! This is a typo from one of the previous versions of this text. It was about the solid model for FEM simulation. Further, we decided not to show it and to limit it to pure simulation outputs in Fig.6-7 and the workpiece-only overview in Fig 8. The noted sentence was removed.
Thank you again!
Comment 5: What parameters are simulated by finite element method? It includes material model parameters (constitutive equation), roll 3D model parameters (such as the invasion of the hydraulic tractories), friction state parameters, etc. In particular, line 145 refers to the use of Częstochowa University of Technology real technical parameters for the radial shear rolling basic FEM-model, but no reference or specific description is given.
Basically, the most influential parameters for simulation are material properties (presented by flow curves for various temperatures and various deformation rates for the specified material), the deformation instrument geometry (3D drawing of the real rolling mill), and all of the interaction parameters (instrument movements and speed, friction, initial temperature, and heat transfer). The specified rolling mill is also used for noted reference work [29-30]. These references by the specified text frame were added.
The mill was produced by a small-batch party with individual design features for each item. This is not a widely used serial device and due to this, we didn’t mark it by its own source. It is generally absent strictly speaking.
Comment 6: The radial nonuniformity characteristics of the equivalent strain field can be observed in the equivalent strain evolution during radial-shear rolling shown in Figure 7. How does this come about?
These phenomena are causated by the described in Reference 31 metal flow trajectory and speed features. Some of these features can be illustrated in Fig 1 and by the corresponding textual explanation downthere.
This is why the most intense shear deformations (we can see in the Fig 7 graphs and visually in Fig 6 crossection) in the sliding lines intersection zone are localized. Each small trajectory-oriented element of the outer layer is subjected to compression strain along the workpiece radius and compression strain in the flow direction along the helical trajectory, and stretching strain across the helical trajectory. The values and vector of all of its processes have a gradient along the workpiece radius. Metal flow currents have no sharp border. And this fact the additional grain refinement conditions by the strain level increasing are added [31].
Comment 7: Is it possible to analyze and discuss the distribution of iso-effect variation with respect to the comparison between test verification results and simulation results?
The verification results show good convergence of the data obtained during modeling and laboratory experiment. The geometry of the bar, including the shape of the press tie, obtained as a result of simulation, fully corresponds to the results of the experiment. This indicates the same metal flow. In addition, the annular zones of strain distribution over the cross section of the rod correlate with the resulting structure gradient. The greater the deformation value (during simulation), the smaller the grain (during a laboratory experiment). Also, the results of verification should include measurements of the thermal effect from the surface of the bar with the data obtained during the simulation. Model c predicts the temperature on the bar surface with an error of no more than 10%.
We consider that the obtained structure gradient is the most valuable proof for convergence with the same gradient mined by the FEM.
Comment 8: Line 403 "The laboratory experiment and the computer simulation result high convergence demonstrated also." What does this conclusion refer to specifically?
This conclusion was supplemented and highlighted by yellow, the detailed answer is given in the response to the observation (comment 7) and in the text of the article.
Comment 9: Figure 1, Figure 6, and Figure 8 lack the secondary number, and the drawings in Figure 4 and Figure 5 do not conform to the specifications of scientific papers. Figures 9-11 can be combined into one drawing.
The TEM-images of Fig 9 and Fig 10 were combined into one figure. All of the links have been corrected.
Figures 1, 6, and 8 have no secondary caption design because we can`t see the reasons to overload and obstruct the simple schemes (Fig 1 and please compare it with not noted Fig 3 design) or simple logical visual comparisons (Fig 8) for huge textual lumber. In the last case, we consider Figure 8 capture name "Comparison of the shape change of the rolled experimental bar with the previously obtained FEM model" which will allow not confuse the experimental bar photo and its FEM-model image.
Figure 6 has the all of clarification captions in the left-down corner of each strain image highlighted by large size and bold
Please compare all the noted cases of lack of the secondary number with Figure 4, where we put it originally. We consider that the secondary caption should be used in such cases.
Regarding the inconsistency of the results presented in Figures 4 and 5 with the topic of the work, we would like to clarify our position. These figures show the most valuable input data for computer simulation and laboratory rolling experiment. We have detailly discussed general points in Comment 5. We guess that Comment 5 and the current issue could be caused by one misunderstanding and we hope that after the Comment 5 discussion Reviewer will understand us more clearly.
Thus, Figure 4 shows data on the resistance to deformation of the material under study, which was used to simulate and calculate the rolling mode. Figure 5 shows the results of dilatometric studies, thanks to which the heating mode (T=530 °C) of the initial billet for rolling was chosen.
Comment 10: The unit is missing or incorrect (for example, Table1 MM, line 181 μ≈ 25), punctuation is incorrect (for example, line 128 0.5 ÷ 15, line 129 20 ÷ 770)
Thank you for this comment. It has been corrected.
Comment 11: The manuscript requires significant grammatical corrections. The reviewer understands that this may not be the author’s native language, so I do not expect the grammar to be perfect. However, some of the major points within the manuscript are conflated by grammatical issues.
Thank you for this remark and your kind understanding. The text has been proofread and we performed some corrections.
Reviewer 3 Report
Thanks for your invitation, this paper (materials-2047696) is recommended for publication in Materials in my opinion, and some suggestion have been given as the followings.
This work shows the radial shear rolling applicability for Zr-2.5%Nb alloy deformation for the UFG structure produce through a combination of simulation and experiment. This paper is interesting and well organized. In my opinion, this work is recommended for publication in Materials. Before publication, some questions should be noted.
1. The ratio of length to diameter of the compressive samples should be given, and what is the standard for the ratio? The yield strength is suggested to given,
2.What is the meaning of 0.5÷15 s-1 in line 128, and also 20÷770℃ in line 129? Does “÷” mean to?
3. What is the reason for the changes between the deformation resistance and temperature or strain? The reason is not shown in line 237.
4. How many times have been tested for the hardness of each point? Figure 13 is suggested to give the error bar.
5. There are some minor grammar or spelling errors should be corrected before publication, for example "To analyses…" in line 283 . Change analyses to analyse.
Author Response
Dear Reviewer,
Thank you for your time, consideration, and comments.
1. The ratio of length to diameter of the compressive samples should be given, and what is the standard for the ratio? The yield strength is suggested to given
The ratio between the length and diameter of the sample is h/d=1.2. Increasing this ratio on zirconium alloys is undesirable since it leads to sample collapse and loss of stability during upsetting. This is also recommended by Gleeble and our experience
2. What is the meaning of 0.5÷15 s-1 in line 128, and also 20÷770℃ in line 129? Does “÷” mean to?
It has been corrected 0.5-15 s-1 in line 128, and also 20-770℃ in line 129
3. What is the reason for the changes between the deformation resistance and temperature or strain? The reason is not shown in line 237.
The paragraph has been corrected. The reason is in the next text lines.
It can be observed that the deformation resistance decreased by approximately 80% as a result of the change in temperature from 20 °C to 770 °C, whereas the opposite phenomenon occurred due to an increase in the rate of deformation from 0.5 to 15 s-1. At a temperature of 20 °C the increase in deformation resistance is no more than 5%. And at 770 °C, the increase in resistance to deformation is approximate ~10%. This difference in the influence of the strain rate can be explained by the thermal effect of plastic deformation. At temperatures close to room temperature, it is higher than at higher deformation temperatures.
4. How many times have been tested for the hardness of each point? Figure 13 is suggested to give the error bar.
Thank you for this comment. 5 times each point of the microhardness graph was tested and averaged.
The distance between test pins was no less than 2,5 mark dimension for excluding the hardening influence.
The error bar was added.
5. There are some minor grammar or spelling errors should be corrected before publication, for example "To analyses…" in line 283 . Change analyses to analyse.
The correction has been done.
Thank you
Reviewer 4 Report
Here are my comments and suggestions:
General:
To fully characterise the microstructure, the results of observations made using light microscopy or scanning electron microscopy performed at lower magnifications should be presented in the paper. Transmission microscopy gives only local results.
Linguistic revision needed.
Introduction:
The Introduction section needs to be improved. It is written at a high level of generality. Furthermore, avoid references block, for example: ‘[...] The most important structural elements of channel reactors are pressure pipes [4-7], the integrity… […] less than the design period [6–9]...[…] The Zr-2.5Nb alloy is used for pressure pipes of CANDU channel reactors [2-7, 10-11]…[…] The structure and properties changing by severe plastic deformation (SPD) study of metallic materials, including zirconium-based alloys are relevant [19–25]..[...] mechanism detailed described in [26–29] sources.[…] As these do not emphasise the particular aspects of the cited papers. In particular, when citations are made about specific technical aspects, single / double references, e.g. [1, 2] are encouraged. It is strongly suggested that the references make in-depth comments on the content of the cited papers.
Materials, methods and equipment
Page 3, line 128; page 4, line 142; the unit of strain rate should be written correctly, i.e., with superscript.
Page 3, line 135; is: Gleeble 3800 "Therwocouple welder" kit; should be: Gleeble 3800 "Thermocouple welder" kit.
The article should include a detailed description of the methodology used to carry out the EBSD study.
In the article, the authors state: ‘[..] Changes and structure gradient by sample section EBSD mapping with 2 mm resolution were studied”. Should it not rather be 2 micrometres?
Results and Discussion
Page 4,line: 227, 232, 236; the unit of strain rate should be written correctly, i.e. with superscript.
included.
Figure 9, 10: The following diffractions need to be solved.
What is the percentage of small and large disorientation angles in the sample locations studied - such a result could be obtained from the EBSD data. The result would be valuable and would significantly increase the value of the publication.
The Authors state: […] The axial (central) zone structure presented on Figure 10 has also significance changed. Instead of large randomly orientated grains, after the radial shear rolling processing, the mixture of long narrow elongated strongly deformed grains was formed […]. The microstructure shown in Figure 10 shows that elongated grains/grains free of dislocation are visible in the central part of the sample. The authors are expected to develop a description of the microstructure that takes into account the deformation pattern of the process.
“The grains orientation here the same and correspond to the rolling direction” - How do you know this, after all, the orientation was not determined. Please comment.
The authors did an EBSD study, I do not understand why the paper does not include full-size images, only excerpts in Figure 12.
Fig. 13 - error bars should be included in the fig. showing changes in hardness, additionally the microhardness notation should be corrected, the notation states the load value.
Is a material characterised by such a heterogeneous cross-sectional microstructure and varying levels of properties beneficial from the point of view of its applicability? Could changing the parameters of the deformation process somehow improve the quality of the product?
An analysis of the results obtained should be developed.
Conclusions:
Conclusions should be reformulated and the most important results/achievements corresponding to the title and purpose of the article should be presented, preferably in the form of points.
Author Response
General:
To fully characterise the microstructure, the results of observations made using light microscopy or scanning electron microscopy performed at lower magnifications should be presented in the paper. Transmission microscopy gives only local results.
Linguistic revision needed.
Introduction:
The Introduction section needs to be improved. It is written at a high level of generality. Furthermore, avoid references block, for example: ‘[...] The most important structural elements of channel reactors are pressure pipes [4-7], the integrity… […] less than the design period [6–9]...[…] The Zr-2.5Nb alloy is used for pressure pipes of CANDU channel reactors [2-7, 10-11]…[…] The structure and properties changing by severe plastic deformation (SPD) study of metallic materials, including zirconium-based alloys are relevant [19–25]..[...] mechanism detailed described in [26–29] sources.[…] As these do not emphasise the particular aspects of the cited papers. In particular, when citations are made about specific technical aspects, single / double references, e.g. [1, 2] are encouraged. It is strongly suggested that the references make in-depth comments on the content of the cited papers.
The team of authors would like to discuss the position of the Reviewer in point of the Introduction section. The article's subject is focused on the study of the influence of the process of radial shear rolling on the structure and properties of the zirconium alloy. The relevance of the application of the process of radial shear rolling is described in detail in the section. At the same time, the relevance of the work on the zirconium alloy is described as concisely as possible, indicating the literature necessary and sufficient for understanding the zirconium problems in terms of the issues of Zr-2.5%Nb, in our opinion.
This is not a review article, we cannot analyze each source in detail for dozens of pages. To get a deeper understanding of the problem, we set aside the most relevant verified sources that we ourselves use.
Materials, methods and equipment
Page 3, line 128; page 4, line 142; the unit of strain rate should be written correctly, i.e., with superscript.
Thank you. The issue has been corrected
Page 3, line 135; is: Gleeble 3800 "Therwocouple welder" kit; should be: Gleeble 3800 "Thermocouple welder" kit.
Thank you. The issue has been corrected
The article should include a detailed description of the methodology used to carry out the EBSD study.
In the article, the authors state: ‘[..] Changes and structure gradient by sample section EBSD mapping with 2 mm resolution were studied”. Should it not rather be 2 micrometres?
Thank you for the comment. We mean here the general resolution for EBSD characterization of the sample in general.
The distance between neighbor EBSD Map, between the characterization points for gradient structure examination. Everything is correct, 2 mm.
Results and Discussion
Page 4,line: 227, 232, 236; the unit of strain rate should be written correctly, i.e. with superscript.
included.
Thank you. The issue has been corrected
Figure 9, 10: The following diffractions need to be solved.
The SAED in our case carries the information about dominant grain orientation, not about phase composition differences, and it is clearly described in the corresponding text frame. The SAED pattern rings-looks shape in Fig. 9 (left) means the equiaxed stochastic-oriented structure. The SAED pattern smoothed short lines shape looks in Fig. 9 (right) means the rolling direction-oriented texture. We have homogeneous material and have no phase transition, inclusions, its evolution, or something other which requires the SAED indexes and their description. We would not like to overload and obstruct the simple visual ideas for informless textual lumber. The images - are the face of the work and they should be simple, laconic, understandable and carry a pure, clear idea.
What is the percentage of small and large disorientation angles in the sample locations studied - such a result could be obtained from the EBSD data. The result would be valuable and would significantly increase the value of the publication. The Authors state: […]
Thank you for this comment. This information really will be useful. We will include the original disorientation graphs files to supplementary materials.
The axial (central) zone structure presented on Figure 10 has also significance changed. Instead of large randomly orientated grains, after the radial shear rolling processing, the mixture of long narrow elongated strongly deformed grains was formed […]. The microstructure shown in Figure 10 shows that elongated grains/grains free of dislocation are visible in the central part of the sample. The authors are expected to develop a description of the microstructure that takes into account the deformation pattern of the process.
Thank you for this comment. We also catch this issue and consider this effect is causated by the large deformational heating effect for the central area mixed with hindered heat transfer from the specified area. Probably it can be the reason for the recrystallization of some grains in the most stressed points.
“The grains orientation here the same and correspond to the rolling direction” - How do you know this, after all, the orientation was not determined. Please comment.
SAED in the right-upper BF TEM-image corner clearly shows the predominant orientation existence.
Also, the same structure type in EBSD from the corresponding central zone is elongated along the rolling direction too. And it also has the predominant orientation which is clearly illustrated by the EBSD miniatures under the graph according to the IPF coloring mark on the figure legend. The central zone grains are elongated along the rolling direction as normal everywhere, not across.
The authors did an EBSD study, I do not understand why the paper does not include full-size images, only excerpts in Figure 12.
Thank you. We will include the all of original EBSD image files to supplementary materials.
Fig. 13 - error bars should be included in the fig. showing changes in hardness, additionally the microhardness notation should be corrected, the notation states the load value.
Thank you. The error bars marks have been added
Is a material characterised by such a heterogeneous cross-sectional microstructure and varying levels of properties beneficial from the point of view of its applicability?
The ultrafine-grained recrystallized structure in the near-surface layers of the finished product can be of practical importance. Since it is she who primarily interacts with an aggressive environment during operation. This structure should make it possible to increase the corrosion resistance of the structural material.
Could changing the parameters of the deformation process somehow improve the quality of the product?
An analysis of the results obtained should be developed.
As part of the work, the conditions for obtaining an ultrafine-grained structure by an industrial (not laboratory) method were determined. The metal after deformation-heat treatment differs from the existing technology in the presence of an ultrafine-grained structure.
An analysis of the results should be developed.
Conclusions:
Conclusions should be reformulated and the most important results/achievements corresponding to the title and purpose of the article should be presented, preferably in the form of points.
Thank you for this comment. We suggest the next additional text to the Conclusion.
It is noted that the rolling temperature should not exceed T = 530 °C in order for the deformation process to take place advantageously in the single-phase α-region. It is advisable to give single reductions per pass in the range from 10 to 25% in order to provide an ultrafine-grained structure in the near-surface layers and prevent metal destruction.
Round 2
Reviewer 2 Report
None.
Author Response
Dear Reviewer,
Thank you for your additional consideration.
best regards, authors' team
Reviewer 4 Report
Some of the comments have not been well addressed.
General:
To fully characterise the microstructure, the results of observations made using light microscopy or scanning electron microscopy performed at lower magnifications should be presented in the paper. Transmission microscopy gives only local results.
- You did not comment on this.
Introduction
Thank you very much for your reply, but I cannot quite agree with the explanations. The introduction section at the moment is written at a high level of generality, and it is important to give specific data. For example this sentence: “The structure and properties changing by severe plastic deformation (SPD) study of metallic materials, including zirconium-based alloys are relevant [19–25”]. It would be useful to develop the thought and broaden the description. One should not generalise; this is after all a scientific paper. Please try to modify this part of the article a bit, it is not practised to block out literature items as much as previously written max. At the moment, the authors refer to, e.g. six literature items in one sentence.
Materials, methods and equipment
The SEM/EBSD methos should be described in detail:
What accelerating voltage was used for the EBSD tests/tilt angle/distance from the column/ what step was used to obtain maps of the crystal lattice orientation distribution / which magnification??
Results and Discussion
I don't quite agree with the answer regarding SAED, but I accept it.
Author Response
Dear Reviewer,
Thank you for your additional consideration.
General:
Sorry, I remember this point, but probably we lose it during the final answer compilation. I`m so sorry and we apologize.
Of course, you are right and transmission microscopy gives very local results. All kinds of microscopy are high locality methods. Due to this, the sample characterization by microscopy should be based on systematic and statistics as much as possible.
At once microscopy is the one of most subjective research methods and we should formalize and digitize the outputs. Fewer description words and more digits and mathematics.
And precisely based on these terms we decide to use the systematic EBSD mapping for each 2 mm (6 maps) of sample cross-section radius with statistical grain data collection and its digital description. Not only a couple of TEM images like on most same works.
TEM details for the structure were obtained on the 13.3k and 39.6k magnification scales to show the most common locations. However, for SEM/EBSD structure characterization we use 8k magnification just according to your comment. This scale is the best scale for grain structure description through all ample cross-section for all types of structure. This scale gives 40-60 enough detalizated grains per map. This is enough scale for objective structure characterization. We do not use optical microscopy due to its inefficiency for UFG submicron and less grain dimension detection. It has a fundamental resolution limitation and known troubles with grain boundary etching. And we have the more powerful EBSD method, which gives much more characterize information.
Introduction
Thank you for your explanation. We tend to agree with your arguments and change the introduction frames for more detail and less general by changing the sources use policy. The text highlighted in pink
Materials, methods and equipment
Yes, of course, we should be made it earlier. A detailed EBSD tech description was added to the text and highlighted in pink
The EBSD mapping using the next conditions was obtained: accelerating voltage 30kV; work distance (distance from column) 10 mm with sample tilt 70° (pretilted holder); magnification 8K; map resolution 318x234 pixels or 14x10 µm; step size (pixel size) 45.1 nm; scanning time 30 min.
Results and Discussion
Thank you for your kind consideration.
best regards, authors' team